# Made to Measure: Patient-Tailored Treatment of Multiple Sclerosis Using Cell-Based Therapies

**DOI:** 10.3390/ijms22147536

**Published:** 2021-07-14

**Authors:** Inez Wens, Ibo Janssens, Judith Derdelinckx, Megha Meena, Barbara Willekens, Nathalie Cools

**Affiliations:** 1Laboratory of Experimental Hematology, Vaccine and Infectious Disease Institute (Vaxinfectio), University of Antwerp, Middelheimlaan 1, B-2020 Antwerpen, Belgium; Ibo.Janssens@uantwerpen.be (I.J.); Judith.Derdelinckx@uza.be (J.D.); meghameena259@gmail.com (M.M.); barbara.willekens@uza.be (B.W.); Nathalie.Cools@uantwerpen.be (N.C.); 2Department of Neurology, Antwerp University Hospital, Drie Eikenstraat 655, B-2650 Edegem, Belgium; 3Center for Cell Therapy and Regenerative Medicine (CCRG), Antwerp University Hospital, Drie Eikenstraat 655, B-2650 Edegem, Belgium

**Keywords:** multiple sclerosis, cell-based therapy, tolerance

## Abstract

Currently, there is still no cure for multiple sclerosis (MS), which is an autoimmune and neurodegenerative disease of the central nervous system. Treatment options predominantly consist of drugs that affect adaptive immunity and lead to a reduction of the inflammatory disease activity. A broad range of possible cell-based therapeutic options are being explored in the treatment of autoimmune diseases, including MS. This review aims to provide an overview of recent and future advances in the development of cell-based treatment options for the induction of tolerance in MS. Here, we will focus on haematopoietic stem cells, mesenchymal stromal cells, regulatory T cells and dendritic cells. We will also focus on less familiar cell types that are used in cell therapy, including B cells, natural killer cells and peripheral blood mononuclear cells. We will address key issues regarding the depicted therapies and highlight the major challenges that lie ahead to successfully reverse autoimmune diseases, such as MS, while minimising the side effects. Although cell-based therapies are well known and used in the treatment of several cancers, cell-based treatment options hold promise for the future treatment of autoimmune diseases in general, and MS in particular.

## 1. Introduction 

Multiple sclerosis (MS) is an autoimmune and degenerative disease of the central nervous system (CNS) that is characterised by demyelination, axonal degeneration and gliosis [1]. It is the leading cause of non-traumatic neurological disability in young and middle-aged adults [2,3]; approximately 500,000 people in Europe and 2.5 million people worldwide have been diagnosed with MS. Intra- and interindividual heterogeneity in the presentation and evolution of the disease are common among patients with MS. According to Lublin et al., there are four disease courses of MS [4]. Relapsing–remitting MS (RRMS) is characterised by inflammatory attacks resulting in new or increasing neurologic symptoms (relapses) followed by periods of partial or complete recovery (remission). Secondary progressive MS (SPMS), which usually occurs within 15 years in approximately half of individuals with RRMS without treatment, is characterised by a continuous, irreversible neurological decline, reduction in brain volume and axonal loss. A primary progressive disease course (PPMS) is present in 10–15% of patients with MS and involves a steady worsening of symptoms with no periodic relapses or remissions. Progressive-relapsing MS (PRMS) is a rare form of MS that is aggressive at onset and involves frequent attacks without recovery of symptoms [1,5,6,7,8].

MS is a complex disease that results from a combination of genetic burden of several risk alleles and environmental factors, including low vitamin D levels, smoking and previous infection with Epstein–Barr virus, affecting immune homeostasis in MS patients [1]. Although the aetiology of MS remains to be elucidated, a dysregulated T cell immune response that involves T helper type 1 (Th1) and Th17 lymphocytes forms the pathophysiological basis of this autoimmune disease [5,9,10,11,12]. Hyperactive dendritic cells (DCs), proinflammatory T and B cells, and functionally impaired regulatory T cells (Tregs) are believed to contribute to the pathogenesis of the disease [13]. A significantly high number of peripheral immune cells enters the CNS through the disrupted blood–brain barrier (BBB), the blood–cerebrospinal fluid (CSF) barrier or the subarachnoid space [14], which results in the attack of self-antigens, including myelin-derived proteins and unknown antigens, in the CNS [15,16,17,18,19].

The strong involvement of the adaptive immune system in the pathophysiology of MS has given rise to the development of therapies that focus on immune modulation and are directed at reducing inflammation, thereby limiting neuronal and axonal damage. Currently, several disease-modifying treatments (DMTs) have been approved by regulatory authorities and are available for the treatment of MS. All DMTs, including classic injectable drugs, new oral substances, and monoclonal antibodies, are characterised by an immunomodulatory action, as reviewed by Hauser and Cree [20]. Although substantial progress has been made with current therapies, efficacy is associated with an increased risk of side effects. Indeed, treatment-related side effects or risks can be severe, including cardiac dysfunction, increased risk of autoimmune diseases and increased risk of infections and cancer [14,21]. Another drawback is that typically, current treatment modalities require life-long therapy, since they do not restore immune tolerance to the self-antigens targeted by the autoreactive immune response in MS patients. Therefore, cell-based therapy may provide an alternative or adjunctive approach. It is envisaged that cell-based therapy has the potential to provide a personalised and effective treatment option that lowers morbidity by uniting efficacy with reduced occurrence of side effects and less frequent hospitalisations, enhancing the quality of life of patients.

To date, various cell types have been investigated to determine if they establish long-term immune tolerance in MS (Figure 1). Some cell-based therapies aim to selectively restore the failed immune tolerance, which is the case for regulatory T cells (Tregs) and tolerogenic dendritic cells (tolDCs), while other cell-based therapies will be used for immune reconstitution after generalised strong immunosuppression, for instance following hematopoietic stem cell transplantation (HSCT). In this paper, we provide an overview of recent and future developments in cell-based therapy.

## 2. Cell Therapy Approaches for the Treatment of Multiple Sclerosis 

### 2.1. Haematopoietic Stem Cells 

Haematopoietic stem cells (HSCs) expressing CD34^+^CD38^–^CD90^+^CD45RA^–^CD49f^+^ are immature pluripotent cells that can develop into all types of blood cells of both the lymphoid and myeloid lineages [22]. Hence, the aim of HSC transplantation (HSCT) is to give a one-time treatment that provides long-lasting disease stabilisation. Indeed, following immunoablation with immunosuppressive drugs, which are used to eliminate all pathogenic autoreactive lymphocytes and reduce inflammation in the CNS, patients are treated with HSCT to support haematopoiesis, thereby renewing the immune system and restoring self-tolerance. HSCs can be isolated from the bone marrow or peripheral blood after mobilisation with drugs, such as cyclophosphamide and/or granulocyte-colony stimulating factor (G-CSF), that enhance proliferation of HSCs and drive them from the bone marrow into the peripheral blood [23,24]. The following recommendations are made regarding collection of HSCs, according to the handbook of the European Society for Blood and Marrow Transplantation (EBMT) [25]. Bone marrow is the preferred source of HSCs. Multiple bone marrow aspirations of 5 mL each, with a maximum of 20 mL/kg donor bodyweight, are suggested to acquire a target dose of 3 × 10^6^ CD34^+^ cells/kg. However, peripheral blood stem cell collection is favoured, as it is considered as less stressful for the patient and leads to faster engraftment and hematologic reconstitution. For this, the minimum target is 2 × 10^6^ CD34^+^ cells/kg collected by leukapheresis. However, higher amounts of cells are aimed for, namely 4–5 × 10^6^ CD34^+^ cells/kg, resulting in faster neutrophil and platelet recovery, and reduced hospitalisation, blood transfusion and antibiotic therapy. Autoimmune diseases, such as MS, are generally treated with autologous HSCT (AHSCT) for safety reasons [26]. 

At the clinical level, AHSCT is a rescue treatment in young patients with RRMS who have low or medium disability grades due to an aggressive inflammatory disease course and in whom other highly efficacious treatments have failed [27,28,29]. One study showed that the proportion of patients with MS who achieved no evidence of disease activity (NEDA) after AHSCT was very high compared with patients who received approved DMTs [28]. In a recent meta-analysis [30], 83% of patients who received AHSCT showed NEDA after 2 years and 67% maintained NEDA after 5 years. The main risk associated with AHSCT is treatment-related mortality, albeit that this risk has decreased from 3.6% to 0.3% in patients transplanted after 2005 [28]. A recent study from the Italian bone marrow transplantation (BMT)-MS Study Group reported that there were no deaths in patients transplanted after 2007 [31]. In a cohort of 210 MS patients with a median baseline expanded disability status scale (EDSS) of 6.0, a high proportion had a durable disease remission up to 5–10 years after the procedure; some of these patients had progressive MS [31]. The Swedish Board of Health and Welfare considers AHSCT as a valid treatment option for patients with active MS [29,32] and several consensus recommendations for the use of AHSCT in MS have been published in the past few years [26,33,34,35]. However, for patients with very advanced MS and high levels of disability, HSCT can neither reverse nor stop the progression of the disease [36] and, therefore, is not recommended. 

Until recently, most studies on AHSCT were observational or prospective single-arm clinical trials (Table 1) [37,38,39,40,41,42]. In one randomised controlled trial, the researchers compared AHSCT with treatment with mitoxantrone, which is rarely used to treat MS today [38]. The MS International Stem Cell Transplant (MIST)-trial (NCT00273364) demonstrated the superiority of AHSCT versus DMTs in terms of the time to disease progression [43]. More recently, an observational cohort study compared outcomes after treatment with alemtuzumab and AHSCT and found that the chance of maintaining NEDA was significantly higher in the AHSCT-treated group [44]. Several clinical trials, comparing the effects of AHSCT with high efficacy DMTs in patients with active RRMS, are ongoing (Table 1). These include the BEAT-MS (NCT04047628), RAM-MS (NCT03477500) and STAR-MS (ISRCTN88667898) [27]. These trials will determine the comparative efficacy of AHSCT and currently available and highly efficacious DMTs, such as alemtuzumab, natalizumab and ocrelizumab. 

The immunological effects that underlie the radical change in the disease course of MS following AHSCT are only partially understood. It has been observed that natural killer (NK) cells, CD8^+^ T cells and B cells repopulate within weeks to months, whereas the reconstitution of CD4^+^ T cells can take up to 2 years [62,63,64,65,66,67,68]. T cells generated after AHSCT undergo selection and maturation in the thymus and show a more diverse profile with new T cell receptor (TCR) clones compared with the dominant clones that were present before AHSCT and that were largely removed by the immunoablative treatment before the transplantation. It has been shown that more than 90% of pre-existing T cell clones are removed from the peripheral blood and the CSF and replaced with clonotypes from the graft [62]. This is predominantly the case for CD4^+^ T cells and, to a much lesser extent, for CD8^+^ T cells [63,64,65]. Whether this limited depletion of CD8^+^ T cells is associated with relapses or disease progression after AHSCT remains to be determined. In this context, mucosal-associated invariant T (MAIT) cells, a novel CD161^high^CD8^+^ cell population originating in the gut mucosa but expressing the CNS-homing receptor CCR6, have been found in lesions in the brains of patients with MS [65]. Nonetheless, myelin-specific T cells are still found after AHSCT, albeit with a strongly reduced capacity to differentiate into Th17 cells compared with their ability prior to the transplantation [67]. Interestingly, changes in the gene expression profiles of CD4^+^ and CD8^+^ T cells have been described, which suggests that the gene expression normalises in CD8^+^ T cells after AHSCT [68]. The rapid reconstitution of NK cells contributes to the suppression of Th17 cell reconstitution [66]; immune regulation is further enhanced by the expansion of Tregs [69]. Furthermore, although all B cells, except for plasma cells, are depleted during HSCT, one study demonstrated that oligoclonal bands persist after the transplantation, which suggests that immunoglobulin-producing cells are not depleted or are insufficiently depleted in the CNS [36]. This observation has been challenged by Larsson et al. [70] who showed that intrathecal immunoglobulin production and neurofilament light levels were lower after HSCT treatment and further decreased over time. Whereas differences in patient characteristics, such as disease duration, disease type, and disease heterogeneity, or treatment-related factors such as the conditioning regimen, may underly the observed discrepancies, studies involving larger cohorts as well as investigating the mechanisms of B cell reconstitution after HSCT are needed.

In conclusion, HSCT can be a treatment option in select young patients with aggressive RRMS who failed to respond to DMTs [26,27,71]. Immunological changes that occur after HSCT in MS are suggestive of long-term induction of immune tolerance. To date, no cellular biomarkers have been identified that can predict which patients will benefit most from this procedure. 

### 2.2. Mesenchymal Stromal Cells 

Mesenchymal stromal cells (MSCs) are multipotent cells that have the ability of self-renewal; MSCs can differentiate into various tissues of mesodermal origin, such as osteocytes, chondrocytes and adipocytes, and other embryonic lineages. MSCs are characterised by the expression of CD73, CD105 and CD90 and the absence of expression of haematopoietic markers (i.e., CD45, CD34 and HLA-DR) and vascular markers (i.e., CD31) markers [72,73]. Given their adult cell potency, MSCs are often called mesenchymal stem cells, although they are more accurately called multipotent mesenchymal stromal cells. MSCs were first described in the 1960s by Friedenstein who isolated them from rodent bone marrow through their inherent adherence to plastic [74]. Currently, MSCs can be isolated from blood, bone marrow, skeletal muscle, adipose tissue, synovial membranes, and other connective tissues. Regardless of the isolation procedure, quantities of MSC obtained from primary tissues are not sufficient for any application in clinical settings. Hence, in vitro propagation is almost always required to achieve a sufficient cell number for in vivo application. MSCs have generated great interest because of their therapeutic ability to induce a profound immunosuppressive and anti-inflammatory effect in vitro and in vivo [75]. The mechanisms by which MSCs exert their immunosuppressive effect are not completely understood. It is thought that they change the inflammatory environment into an anti-inflammatory environment directly by paracrine signals and by several secreted soluble factors, such as transforming growth factor beta (TGF-β) [76], hepatocyte growth factor [76], indoleamine 2,3-dioxygenase (IDO) [77], nitric oxide [78], interleukin (IL)-10 [79] and prostaglandin E2 [80], and through cell-to-cell contact via the inhibitory molecule programmed death 1 (PD-1) [81]. MSCs also work indirectly via the recruitment of other regulatory networks that involve antigen-presenting cells (APCs) [82] and Tregs [83]. However, it is evident that MSC-induced unresponsiveness lacks any selectivity. MSCs mainly affect the functions of T cells; for instance, MSCs induce a cell cycle arrest in anergic T cells or a cytokine profile shift in the Th1/Th2 balance towards the anti-inflammatory Th2 phenotype [84,85]. Furthermore, MSCs suppress the cytolytic effects of cytotoxic T cells [86]. MSCs are also capable of inhibiting NK cells [87,88], B cells and APCs. Furthermore, MSCs have been reported to promote the formation of potent CD4^+^CD25^+^ and CD8^+^ Tregs in vitro and in vivo [83,89]. 

Several phase I and II clinical trials used MSCs derived from allogeneic donors and evaluated their effect on autoimmune diseases, including type 1 diabetes (T1D), rheumatoid arthritis (RA) and MS (Table 1) [90]. Since MSCs represent only a small fraction (0.001–0.01%) of total nucleated cells in bone marrow and other tissues, it was mandatory for these studies that the MSCs were expanded ex vivo from a small bone marrow aspirate under clinical-grade conditions to significant numbers in 8–10 weeks [91,92]. Most of the reported trials, to date, were uncontrolled open-label phase I studies including patients with RR-MS, SP-MS, and PP-MS. A review of trials found that MSCs were safe and tolerated by patients with MS [93]. More recently, a randomised placebo-controlled phase II clinical trial found that five out of nine patients with MS who received an intravenous infusion of bone marrow-derived MSCs had a trend to lower cumulative numbers of gadolinium-enhancing lesions at 6 months following infusion, as shown by magnetic resonance imaging (MRI) [48]. However, there was no significant decrease in the frequency of Th1 cells in the peripheral blood of patients treated with MSCs. Interestingly, MSCs are likely to promote neuroprotection in addition to their immunomodulatory characteristics [94,95,96]. Indeed, MSCs could promote endogenous repair by recruiting local neural precursor cells, possibly through the secretion of neurotrophic factors, thereby driving neurogenesis and remyelination [97,98]. The migratory potential and homing capacity of these cells into the CNS still needs to be clarified.

The clinical results obtained using MSC therapy in patients with MS confirmed the feasibility and safety of an in vivo application of MSC without major adverse events. However, the migratory potential and homing capacity of these cells into the CNS as well as the clinical significance of these findings need to be corroborated. 

### 2.3. Regulatory T Cells 

Tregs are a subset of CD4^+^ T cells that play an important role in the balance between immunity and tolerance. These cells are characterised by the expression of high levels of IL-2 receptor α chain (IL-2Rα/CD25) and Forkhead box P3 (FoxP3) [99], which is a master regulator that orchestrates the transcriptional machinery that induces Treg-relevant genes, such as il2ra (CD25) and ctla-4, by binding over 1400 genes and acting as a transcriptional repressor and activator [100,101,102]. Its expression is inversely correlated with the expression of IL-7R (CD127) [103]. FoxP3 Tregs are generally subdivided into thymic-derived or naturally occurring Tregs (nTregs) and peripheral-induced Tregs (iTregs), which have phenotypic and functional similarities, as well as differences in stability and gene expression [99,104]. It is currently accepted that nTregs control immune responses to self-antigens, while iTregs suppress inflammation at mucosal barriers [105]. A current study defined Tregs as a heterogenous mixture of cellular sub-phenotypes with a high degree of phenotypic complexity that reflected different states of maturation, differentiation and activation [106].

Tregs are responsible for minimising the damage to the body’s own cells and tissues during persistent immunity and for maintaining tolerance to self. For this, Tregs act predominantly by suppressing, eliminating, or inactivating effector T cells, including autoreactive T cells, in the periphery [99,107]. Consequently, it is believed that the disruption of Treg numbers and/or function gives free rein to self-reactive T cells, which may contribute to an increased susceptibility to autoimmune diseases [108]. Indeed, reduced numbers or the impaired functionality of Tregs have been associated with the development of different autoimmune diseases, including MS [5], RA [109], T1D [110], psoriasis [111], myasthenia gravis [112] and autoimmune polyglandular syndrome type II [113]. Hence, restoring tolerance in patients with these diseases could be the key to overcoming autoimmunity. In this regard, adoptive cell transfer of Tregs has proven to be effective in preventing autoimmunity [114,115] and graft-versus-host disease (GVHD) [116,117], and in delaying graft rejection in preclinical animal models [118,119]. 

The suppressive repertoire of Tregs involves the secretion of immunosuppressive cytokines, such as IL-10, IL-35 and TGF-β, and cytotoxic molecules, such as granzyme B and perforin, as well as contact-dependent suppression (e.g., CTLA-4). Additionally, Tregs can indirectly affect immune tolerance by suppression of APCs, such as DCs (extensively reviewed in [99]). Furthermore, Tregs can transfer their suppressive activity to conventional CD4^+^ T cells, which is termed infectious tolerance [120]. They create a local tolerogenic environment in which naïve T cells convert into cells with an induced Treg phenotype. These cells are responsible for bystander suppression [121] because they induce tolerance to cells involved in the immune reaction without direct interaction. Hence, adoptive cell transfer of Tregs may not require long-term survival of the administered cells and may be used to alleviate the autoimmune response in diseases where it is directed against different self-antigens.

Currently, there is a broad range of Treg isolation and expansion protocols [122]. For instance, effective isolation methods with high purity and efficient expansion protocols are required to preserve the desired cell characteristics. Although Tregs are present throughout the body, peripheral blood is the most commonly used source of Tregs [123]. However, since Tregs comprise only 5–7% of the CD4^+^ T cells that develop in the thymus and in the periphery [124], in vitro Treg expansion is mandatory following isolation of a highly pure Treg population to generate sufficient cells for clinical application [122]. Molecules, including rapamycin [125,126,127], TGF-β [128] and all-trans retinoic acid (ATRA) [129,130], can be used to boost Treg expansion and stability, while preventing outgrowth of contaminating cells. 

Positive preclinical outcomes, a better understanding of the characteristics of Tregs and the possibility of obtaining enough of these cells have paved the way for more than 50 active and completed clinical studies. These studies have tested the safety, feasibility, and efficacy of adoptive cell transfer of Tregs in the context of both autoimmunity and transplantation [131]. Recently, also in MS, the clinical use of autologous CD4^+^CD25^hi^CD127^−^FoxP3^+^ Tregs was evaluated in a phase I/IIa clinical study [57]. Tregs were administered intravenously or intrathecally in RRMS patients, and the safety of the approach was demonstrated (Table 1). Altogether, studies proved the safety of the clinical use of ex vivo expanded polyclonal Tregs and showed promising results in the delay and prevention of graft rejection and in the treatment of autoimmune responses [132]. 

However, the efficacy was not conclusive and often only modest clinical responses were obtained [133]. This could be, at least in part, due to the use of polyclonal Tregs which collectively target a broad mix of antigens that are not all related to the disease, thereby potentially weakening the clinical effect. This is further confirmed in studies in mice demonstrating limited effect of polyclonal Treg infusion in immunocompetent individuals unless high numbers of Tregs are administered [134,135]. Moreover, the use of polyclonal Tregs could cause a transient risk of generalised immunosuppression [136]. In contrast, Tregs isolated from pancreatic draining lymph nodes or pulsed with pancreatic islet antigen are significantly better at preventing disease onset or curing autoimmune-prone non-obese diabetic (NOD) mice compared with polyclonal Tregs [137,138,139,140,141]. Thus, the use of antigen-specific Tregs could help to achieve improved clinical benefit in cases where the disease-causing antigen is known.

More powerful Treg therapies could be engineered by enhancing Treg antigen-specificity or functionality based on the knowledge gained from T cell therapies in oncology [142]. Most efforts involve introducing transgenic TCRs or chimeric antigen receptors (CARs) into Tregs. Although TCRs and CARs are both synthetic receptors, transgenic TCRs maintain the structure of the native TCR, but are designed for antigen selectivity and high affinity. CARs are synthetic fusion molecules that express the antigen recognition domain of a monoclonal antibody and one or more TCR costimulatory signalling domains [99,124]. Both techniques have been tested in different animal models of autoimmune diseases and transplantation [99]. In MS, pathogenic self-reactive T cells are targeted by murine transgenic Tregs which express an extracellular myelin basic protein (MBP) peptide-bound major histocompatibility complex (MHC) that is linked to an intracellular TCR-chain signalling domain. Subsequently, this interaction mimics physiological TCR-signalling on Tregs, resulting in the activation of transgenic Tregs and in the subsequent secretion of high levels of anti-inflammatory cytokines [143]. Furthermore, adoptive transfer of transgenic Tregs was able to prevent and treat MBP-induced experimental autoimmune encephalomyelitis (EAE) [143,144]. Expanded human Tregs, transduced with an MBP-specific TCR, are able to suppress MBP-specific effector T cells effectively in vitro. These transduced cells ameliorate disease in myelin oligodendrocyte glycoprotein (MOG)-induced EAE, which is indicative of the in vivo effect of bystander suppression mediated by soluble factors [145]. Similarly, converting antigen-specific effector T cells into Tregs through the overexpression of FoxP3 is being investigated [146,147]. In one study, engineered Tregs, overexpressing a MOG-specific CAR in trans with the murine FoxP3 gene, demonstrated their suppressive function in vitro [148]. More recently, reestablishment of Treg functionality in patients with MS was reported following in vitro expansion and MBP-specific TCR transduction of Tregs [149]. 

Further research in Tregs as a cell therapy for MS, and other autoimmune diseases, will undoubtedly provide us with interesting new insights.

### 2.4. Tolerogenic Dendritic Cells 

DCs are the most professional APCs and are the sentinels of our immune system. They capture and process exogenous antigens and self-antigens in peripheral tissues [150,151,152] and present them to other immune cells after migration to the secondary lymphoid organs [150,153]. Subsequently, DCs stimulate naïve T cells, effector T cells, memory T cells and B cells. In doing so, DCs bridge the innate and adaptive immune systems [154] and play an important role in the balance between immunity and tolerance [155,156]. In patients with MS, DCs are abundantly present in brain lesions, and display a pro-inflammatory state with an altered phenotype and/or function compared with healthy controls [157]. Specifically, the DCs of patients with MS show upregulated levels of activation markers, such as CD86, CD80 and HLA-DR, and fail to upregulate programmed death ligand 1 (PD-L1) [158,159,160] compared with their healthy counterparts. Moreover, DCs from patients with MS secrete higher levels of immune-stimulatory cytokines, including IL-12p70, IL-18 and IL-23 [157,161,162], compared with DCs from healthy individuals. These findings underscore a potentially important role for DCs in the pathogenesis of diseases, influencing the effector function of auto-reactive T and B cells [163]. 

Alternatively, deploying the tolerogenic potential of DCs could possibly have a positive impact on the balance between immunity and tolerance in MS. For this, DC function can be directly modulated in vivo before they can be used as an immunotherapeutic tool to treat MS [164], or tolerance-inducing or tolerogenic DCs (tolDC) can be generated in vitro from peripheral blood CD14^+^ monocytes [165]. For the latter, several immunosuppressive biologicals and pharmaceuticals, including corticosteroids, TGF-β, dexamethasone, vitamin D_3_ and cyclosporine have been used. These factors have been demonstrated to modulate the differentiation and function of DCs [166,167,168], as evidenced by the maturation-resistant phenotype, intermediate expression of co-stimulatory molecules, a shift towards anti-inflammatory cytokine production and a reduced capacity to stimulate T cell responses [169,170]. Interestingly, the use of vitamin D_3_ is one of the most widely established approaches, as it has significant immune regulatory properties both in vitro and in vivo [171,172,173,174,175,176,177,178,179,180,181,182,183,184,185,186]. These studies showed amongst others that myelin peptide-loaded tolDC, generated with vitamin D_3_, induced stable antigen-specific hyporesponsiveness in myelin-reactive T cells from MS patients in vitro. 

In addition, to guarantee the efficiency and stability of antigen presentation by DCs, several antigen loading strategies have been developed to induce immune responses [152]. These include (1) the in vivo loading of antigens to circulating DCs in patients [187], (2) different ways of in vitro loading of DCs with antigens [188,189,190,191,192,193,194,195] and (3) DC transfection with mRNA-encoding antigens [196,197,198,199,200,201]. 

Although the use of immune-stimulatory DCs to reinforce immune responses against cancer and infectious diseases has been broadly described in multiple clinical trials [202,203,204,205], the use of tolDC as a treatment strategy for autoimmune disorders is still in its infancy. A limited number of studies have exploited the tolerogenic capacity of DCs to treat patients diagnosed with T1D, RA, Crohn’s disease, MS and Neuromyelitis optica (NMO) [58,156,179,206,207,208,209,210]. In particular, in MS (Table 1), one phase 1b clinical trial reported that the use of tolDC, generated with dexamethasone, was safe and feasible in this patient population [58]. In addition, two single-centre clinical phase I/IIa trials in Antwerp, Belgium (NCT02618902) and Badalona, Spain (NCT02903537) are currently investigating the safety and feasibility of tolDC in patients with MS; the studies are also comparing different modes of tolDC administration, i.e., intradermal and intranodal, respectively [59]. The results from these studies will provide new insights into the use of tolDC as a possible treatment option for MS and other autoimmune diseases.

### 2.5. Other Immune Cells

#### 2.5.1. B Cells

B cells play a pleiotropic role in the induction of immune responses. They contribute to immunity through the production of antibodies, antigen presentation to T cells and the secretion of cytokines. There are different subsets of B cells. For instance, early lineage CD20^+^CD79^+^CD27^+^ B cells function primarily as APCs expressing MHC and costimulatory molecules thereby sustaining T cell-mediated cellular responses, whereas late lineage CD138^+^ mature plasma cells and CD38^+^ plasmablasts secrete antibodies, including auto-antibodies, related to the humoral response [211,212]. The role of B cells in autoimmunity has been underlined by the successful therapeutic effect of B cell depletion with anti-CD20 monoclonal antibodies [213]. Rituximab, a chimeric anti-CD20 monoclonal antibody, has proven to be highly beneficial for patients with certain autoimmune diseases, including RA, MS and T1D. However, while plasma cells and oligoclonal bands in the CSF remain unaffected by anti-CD20 therapies, B cell depletion aggravated the symptoms in some patients, which suggests that B cells also have a protective role in autoimmune pathology [214]. In this context, IL-10-producing regulatory CD1d^+^CD5^+^ B cells were found to be able to downregulate the initiation of autoimmune diseases and the onset or severity of EAE, collagen-induced arthritis, contact hypersensitivity and inflammatory bowel disease [215,216]. Therefore, B cell-mediated regulation of the immune system may be of great interest for the development of new cell-based therapies for immunosuppression in the field of autoimmune diseases. Several preclinical studies used different types of B cells as preventive and therapeutic treatment in EAE, which provided preclinical evidence for tolerance induction [217,218,219,220,221]. The adoptive transfer of splenic IL-10-producing CD1d^hi^CD5^+^ regulatory B cells, so-called B10 cells, isolated from mice treated with anti-CD20 monoclonal antibodies, resulted in limited disease severity when the B10 cells were administered before EAE induction [222,223]. More recently, administration of regulatory B cells (Bregs) also promoted oligodendrogenesis and remyelination in an EAE [224]. To our knowledge, no clinical trials have used B cell-based therapy in patients with MS or other autoimmune diseases to date. 

#### 2.5.2. Natural Killer Cells

Natural killer (NK) cells are innate cytotoxic lymphocytes derived from CD34^+^ haematopoietic progenitor cells which are involved in early defence mechanisms [225,226,227]. Human NK cells can be identified by the molecular marker CD56 in the absence of the expression of CD3, while the combination of the expression of CD56^+^ and CD3^+^ define a mixed population of NK-like T cells (NKT) and antigen-experienced T cells [228]. CD56^bright^ NK cells are mostly present in secondary lymphoid tissue, while large numbers of CD56^dim^ NK cells are found in the bone marrow, blood and spleen [225,229]. NK cells induce apoptosis of their target cells by utilising granzyme B and perforin, and by secreting inflammatory cytokines, such as IFN-γ, upon stimulation with IL-12 or other cytokines, which are released by monocytes, macrophages and/or DCs [225,226,227]. More recently, the generation of trained immunity, i.e., immune memory of the innate immune system, has been described [230]. In this perspective, similar functional properties as the adaptive immune system have been ascribed to NK cells, including the expansion of antigen-specific cells, the generation of long-lasting memory cells that are able to persist after encounter with an antigen, and the possible induction of a boosted secondary recall response. 

In MS, NK cells play a dual role because they have protective and pathogenic properties, as evidenced by the contradictory results obtained in EAE [228,229]. This duality is illustrated by the fact that daclizumab, a humanized anti-CD25 monoclonal antibody, reduces the disease activity in many patients with MS, but has led to severe CNS inflammation in 12 patients worldwide [231]. The beneficial mechanism of action of daclizumab was mediated by the expansion of the CD56^bright^ NK cell population, which led to the killing of activated T cells. Regarding the increased CNS autoimmunity on the other hand, it has been speculated that the mechanisms involved led to a decrease in Tregs [232]. Concerns about—potentially autoimmune—hepatotoxicity resulted in the withdrawal of daclizumab from the market in March 2018 [233,234,235]. 

Albeit that NK cell-based immunotherapy shows promising results in early stage clinical trials in haematological malignancies and solid cancers [236], more fundamental research is needed before NK cell-based therapies can be used in human clinical trials in MS. This includes the identification of a regulatory NK cell subset, the optimal procedures for cell isolation, differentiation and expansion protocols and the administration regimen [237].

#### 2.5.3. Natural Killer T Cells

A T cell subset with regulatory properties that exhibits characteristics of NK cells has been identified in mice and humans (extensively reviewed elsewhere [238,239,240]). These NKT cells are a subset of innate lymphocytes that recognise endogenous or exogenous glycolipids in the context of CD1d molecules expressed by APCs, such as monocytes, DCs and myeloid-derived suppressor cells (MDSCs). Upon antigenic stimulation, NKT cells produce a variety of immunomodulatory cytokines, which endows the cells with potent immunoregulatory properties. Nonetheless, various subtypes of NKT cells may have different effects in the immune system [241]. Importantly, NKT cells in MS were described to have the potential to act as both protective and pathogenic lymphocytes [242]. The role of NKT cells in the pathophysiology of MS needs further clarification before they could be used as a cell-based therapy.

The role of NKT cells and their potential for modulation to increase tolerance towards self-antigens have been investigated in vitro and in animal models of various autoimmune diseases [243,244]. However, impaired NKT cell function in patients with autoimmune diseases could hamper the clinical use of autologous NKT cells, unless in vitro manipulation could change their function. Moreover, NKT cells constitute less than 1% of T cells in the peripheral blood [241]. Hence, in vitro expansion is needed to achieve a sufficient cell number for in vivo application [245].

Although NKT cell-based therapy has been investigated in the field of cancer research [241], there have been no studies in animal models of autoimmune diseases. Deciphering the roles of NKT subsets in tolerance induction, selecting the appropriate NKT cell subset and evaluating the effects on animal models of autoimmune disease will be necessary before these cells can prove their value in phase I clinical trials in humans. 

#### 2.5.4. Myeloid-Derived Suppressor Cells

MDSCs are innate immune cells from the myeloid linage and are important for creating an immunosuppressive environment in tumours [246]. They play a protective role in autoimmune diseases through the inhibition of T cell-mediated immune responses [246]. Two large groups of cells have been described (extensively reviewed in [247,248,249]). In brief, granulocytic or polymorphonuclear MDSCs (PMN-MDSCs) are similar to neutrophils, while monocytic MDSCs (M-MDSCs) are similar to monocytes. A third, less common population of MDSCs has been described in humans, which is called early-stage MDSCs. 

The role of these cells is more complex in autoimmune diseases. Recently, numerical, phenotypical, and functional differences in MDSCs were demonstrated in patients with RRMS and SPMS [250]. Patients with SPMS had a decreased frequency of M-MDSCs and PMN-MDSCs compared with healthy controls, while the frequency of M-MDSCs and PMN-MDSCs was increased in patients with RRMS during relapse as compared with healthy controls. More importantly, M-MDSCs demonstrated the capacity to suppress T cells in patients with RRMS and healthy controls, while these cells promoted autologous T cell proliferation in patients with SPMS [250]. In EAE, the preventive and therapeutic administration of purified antigen-presenting MDSCs led to lower percentages of activated T cells and higher percentages of regulatory B cells, which implied that MDSCs had tolerogenic properties [251]. More research into MS is needed before MDSCs can be investigated as a therapeutic cell product in human clinical trials.

### 2.6. Use of Cells as Carriers of Antigens to Induce Tolerance

#### 2.6.1. Peripheral Blood Mononuclear Cells

An alternative approach for effective immunosuppression in the treatment of autoimmune diseases involves the coupling of self-antigen-derived peptides to cellular vehicles using chemical fixatives [252]. The induction of immunosuppression using this method is indirect and implies that the fixed cells rapidly undergo apoptotic cell death following fixation and subsequently carry over intact peptides to tolerogenic APCs for processing and presentation [253,254]. Lutterotti et al. performed an open-label, single-centre, dose-escalating phase I/IIa study to evaluate the therapeutic use of autologous peripheral blood mononuclear cells (PBMCs) in nine patients with MS: two patients had SPMS and seven patients had RRMS (Table 1). The PBMCs were coupled with seven myelin-derived peptides that were associated with MS pathogenesis and against which demonstrable responses could be detected in the patients included in the trial [60,255]. Administration of the myelin-derived peptide-loaded PBMCs was reported to be feasible, safe and well tolerated. Interestingly, patients who received a high cell dose showed diminished antigen-specific T cell responses [60]. Despite the advantages associated with the limited time for the preparation of the cell product, the correct dose and frequency of the treatment remain unknown. 

#### 2.6.2. Erythrocytes

Erythrocytes, which are also known as red blood cells (RBCs), are the most common type of blood cell. Their main function is to deliver oxygen to body tissues. RBCs are continuously cleared from circulation through phagocytosis without eliciting an autoimmune response. Hence, the tolerogenic properties of these apoptotic cells can be used to engineer tolerance-inducing RBCs. Pishesha et al. described one such technique, called sortagging, sortase-mediated transpeptidation [256]. Engineered RBCs that were covalently linked to MOG_35–55_ protect against and reverse early signs of EAE [256]. A phase Ib clinical trial involving this approach started recruiting patients with MS in October 2017 (Table 1) [257]. Results were presented as a late-breaking abstract during ECTRIMS 2019 [61]. The investigators reported that there was a reduction in antigen-specific T cell responses to myelin peptides in the high-dose group, whereas the proportion of type 1 regulatory T cells (Tr1) and nTregs, and IL-10 levels increased providing evidence of immune tolerance induced by this treatment strategy. 

## 3. Key Issues When Designing Cell-Based Therapies For MS

### 3.1. Autologous Versus Allogeneic Therapy

Cell products for tolerance induction can be derived from the same individual (autologous) or another individual (allogeneic). From a practical point of view, there are many advantages associated with the use of allogeneic cell therapy. For instance, allogeneic cell therapy has a lower production cost compared with the cost related to individualised autologous cell products. There is also a higher availability of allogeneic cell products because cryopreserved stocks can be used, which means that they are available as off-the-shelf products [258]. However, the risk of host immune rejection due to GVHD is substantial in allogeneic cell therapy and requires simultaneous strong immune suppression to allow cell engraftment for immune-modulatory purposes. Autoimmune patients are unlikely to undergo the same heavy lymphodepletion as patients with cancer, which makes it even harder to evade the immune system with an allogeneic product. In contrast, the risk is minimal in autologous therapy. Additionally, donor screening is much stricter for allogeneic cell therapy in terms of infectious screening, such as for (human leukocyte antigens) HLA typing, which results in increased costs [258]. In addition, because most patients with autoimmune diseases do not have the same urgency to begin cell therapy as patients with cancer, apart from a life-threatening flare-up, the benefits of an autologous patient-specific cell therapy product may outweigh the benefits of off-the-shelf therapy in the autoimmune setting. Given these issues, autologous therapy is often preferred over allogeneic therapy for tolerance induction, and its long-term persistence could justify its high price tag. For example, both European and American guidelines do not recommend allogeneic HSCT in patients with MS [259,260]. Moreover, also allogeneic Treg therapy has only been tested in immunosuppressed and immunocompromised individuals [122]. Nonetheless, future design of more universal cell-based therapies could potentially result from more knowledge and research using CRISPR-Cas9 technology to render cells HLA deficient or to induce the ectopic expression of non-canonical HLA-E or HLA-G genes, which are expressed during maternal–foetal tolerance [124].

### 3.2. Antigen-Specificity

General immune modulation may be accompanied by undesired side effects, such as opportunistic infections and secondary autoimmunity. Therefore, harnessing the immune system to restore immune tolerance using tolerance-inducing cell strategies requires loading the cell product with myelin antigens or receptors, depending on the cell type used, to acquire disease-related antigen specificity. 

Although substituting only 15–30% of total myelin content [261], the myelin proteins are presumed to be the major antigenic targets of the MS-driving autoimmune response [262]. The protein content within the myelin sheath is predominantly composed of proteolipid protein and MBP, as well as other myelin proteins, such as MOG [261]. Irrespective of their abundance in the myelin sheath, epitopes from these three myelin proteins have been shown to be encephalitogenic in different animal models [263]. Thus, the reactivity towards a wide variety of myelin peptides can be detected in patients with MS [264,265]. Hence, directing myelin specificity to cell-based therapies for MS may represent a promising approach to tackle MS-related autoimmunity. In this way, the dysregulated myelin-directed immune response could be restored, without affecting the normal surveillance and effector function of the immune system.

To date, however, few clinical trials have investigated myelin-specific cell-based therapies. Indeed, many of the above-mentioned cellular treatments do not have a myelin-specific mode of action, although encouraging safety results have been demonstrated for several antigen-specific treatment approaches in phase I and II clinical trials for MS [58,60,266,267,268,269], including cell-based therapies with DC [58] and mononuclear cells [60].

Nevertheless, various pitfalls have limited the development of antigen-specific treatment. First, even though myelin proteins are suggested to be culprit antigens, no single antigenic target has been identified. Myelin reactivity in patients with MS is heterogeneous and possibly dynamic because of the emergence of neo-autoreactivities due to disease activity-related tissue damage, which is associated with epitope spreading [270,271,272]. Therefore, there is no obvious single peptide or peptide mix at which tolerance reconstitution can be aimed. Moreover, even though ex vivo reactivity can be directed towards a wide variety of myelin peptides, some are non-pathogenic, such as the so-called cryptic or not naturally processed epitopes [273]. These factors complicate the choice of targets for antigen-specific therapy. Nonetheless, few side effects were reported in clinical trials with antigen-specific therapies [16,274]. However, a risk of inducing MS exacerbation or hypersensitivity reactions when trying to modulate the immune system in a myelin-specific way remains. In this context, the administration of myelin antigens by means of carrier cells might represent a more controlled approach to induce stable and antigen-specific immune tolerance.

Several innovative antigen-specific treatment strategies are currently in the preclinical phase and may address some of the previously mentioned issues. New antigen-loading strategies are being investigated as alternatives to classical peptide pulsing. For instance, transfection with viral vectors or nucleic acids encoding full-length myelin proteins may lead to the presentation of a wide variety of naturally processed myelin peptides. These new strategies could be used to increase the efficacy of current cell-based antigen-specific treatment approaches, as well as to add antigen-specificity to cell therapies that are not yet specifically directed towards the myelin response, including MSC-, HSC- and Treg-based strategies. These new approaches may represent an intriguing opportunity for antigen-specific cell treatment. 

### 3.3. Migration Across the Blood–Brain Barrier

The trafficking of cell-based therapies into the CNS can be used for targeted immunotherapy against various neuroinflammatory diseases [275,276,277,278]. Indeed, the triumph of cell-based immunotherapy in inducing immune tolerance depends on the accurate delivery and trafficking of the therapeutic, i.e., tolerance-inducing cells, to the inflammatory sites [279,280]. Hence, a clear understanding of the underlying mechanisms involved in cell migration is necessary to advance the development of new therapies. However, entry into the CNS is heavily restricted by the blood–brain barrier (BBB), a diffusion barrier that tightly regulates homeostasis of the CNS and impedes the influx of most compounds from the blood to the brain [281,282,283]. The restrictive nature of the BBB provides an obstacle for drug delivery to the CNS. Although there have been medical advances in the care of individuals with brain and CNS diseases, the treatment of these disorders remains challenging and insufficient because of the BBB, which prevents many drugs in circulation from reaching the brain. Hence, major efforts have been made in developing methods able to modulate or bypass the BBB for delivery of therapeutics [284]. Nonetheless, several cell types, including MSCs, Tregs and DCs, can migrate in and out of the BBB efficiently, and BBB-transmigratory capacity of the cells could be exploited for the therapeutic targeting of the inflammatory disease mechanism in the CNS. Moreover, these cells, owing to their ease of isolation, established safety and potential to target different pathways in neuronal regeneration [285,286,287], have proved to be attractive therapeutic agents and can secrete various cytokines and growth factors with anti-apoptotic, neuroprotective and immune-modulatory properties [277,288]. They can be used as vehicles to deliver antitumor therapeutics for brain tumour treatment and recent reports have demonstrated that they can interact and migrate across the BBB under injury or inflammation. They express a variety of leukocyte-like homing molecules, such as chemokine receptors and adhesion molecules [289,290,291] and they use a multistep homing cascade (i.e., rolling, adhesion, and transmigration) to engage with endothelial cells [292,293,294]. Indeed, these cells use adhesion molecules, including vascular cell adhesion molecule (VCAM)-1 and β1 integrin, to transmigrate through the endothelial barrier and preferentially transmigrate on TNF-α-activated endothelium rather than naïve endothelium [295,296]. Several chemokine receptors and their ligands, including CXCL9, CXCL16, CCL20 and CCL25, are known to be explicitly involved in the cell transmigration through the endothelial layer [285,295,296,297,298]. 

Although in general, these cells undertake the same migratory cascade to reach the CNS by moving across the BBB, they still require specific mechanisms for their mode of action. For instance, MSCs favourably transmigrate through the endothelial cells using G-protein-coupled receptor signalling-(GPCR-) dependent pathways [285]. MSCs migrate either by paracellular or transcellular diapedesis through discrete gaps or pores in the endothelial monolayer that are enriched for VCAM-1 (transmigratory cups). In contrast to leukocytes, their transmigration does not involve significant lateral crawling, presumably due to the lack of Mac-1 expression [289]. Similarly, Tregs tend to migrate across the brain endothelium and to suppress the effector T cell functions at the site of emerging inflammation. Recent studies have suggested that the detection of low numbers of Tregs in the CNS of patients with MS [299,300,301] and murine Tregs showed augmented migratory capacity in vitro and in vivo via the BBB [300,301]. In addition, human FoxP3^+^ Tregs migrate across in vitro human brain endothelium at higher rates than other cells. Tregs from patients with RRMS showed impaired migratory abilities in crossing the BBB under non-inflammatory conditions [293]. The integrin CD62L is a crucial lymphoid homing molecule for immune cells and is also an important migration associated molecule for Tregs [302]. The migratory capacity of Tregs, through the BBB is controlled by distinct signals from chemokines/chemokine receptors, such as CCR7 and CCR6 [296]. Additionally, DCs found within the CNS correlate with the severity of disease, and they exhibit more efficient transmigration than the T cells in in vitro models of the BBB [275]. Different chemokine receptors and ligands, including CCR5, are involved in the inflammatory migration of DCs [303] and, thus, should be targeted for the development of therapies. Crossing the BBB is a prerequisite for all these cells to exert their therapeutic effects in treating neurological diseases or CNS injury and is necessary for their use as vehicles for drug delivery to treat brain tumours. Hence, the selective targeting of the trafficking and compartmentalisation of these cell types into different sites to exert their apposite immune suppression would be therapeutically beneficial. In this regard, efforts have been made to increase CNS migratory capacity of cells, such as CCR5-encoding mRNA-electroporation of tolDC. Accordingly, the capacity of mRNA-electroporated tolDCs to transmigrate toward a chemokine gradient in an in vitro model of the BBB improved significantly, and neither the tolerogenic phenotype nor the T cell-stimulatory function of tolDCs was affected [304].

Furthermore, the ability to monitor the migration and fate of these cells under in vivo conditions is helpful in devising rational therapeutic strategies and is also critical for optimisation of these strategies. For this, some non-invasive in vivo cell tracking techniques are used such as in vivo bioluminescence imaging [305]. This is an indirect cell labelling technique with reporter genes which allows cell tracking in small animal models. The mobility of the cells, including MSCs, DCs and Tregs, to the target tissue can be easily verified using in vivo bioluminescence imaging reporter gene strategies as well [306,307,308]. 

There is a need for the ongoing and future clinical studies to focus on the use of various therapeutic strategies that exploit the migration-associated molecules for various cell types [47,309,310,311]. The majority of current clinical studies use intradermal or subcutaneous routes of administration with different outcomes [312,313]. Based on these reports, the effect of the administration route on the efficiency of the therapeutic vaccine remains unclear and a topic of debate. Further optimisation is required to enhance the overall vaccine outcome.

## 4. Discussion and Conclusions

Effective treatment of MS should target the causative mechanisms of disease and induce long-lasting effects. As immune-mediated demyelination and axonal degeneration are essential components of the neurodegenerative process of the disease, the ideal treatment for MS would convert the function of B and T lymphocytes from disease-causing to disease-regulating, without affecting the rest of the immune system. In this context, several cell-based, tolerance-inducing therapies have been developed, including MSCs, Tregs and DCs. 

While translating cell-based therapies from the bench to the bedside, several challenges arise: manufacturing of the cell-product, administration route and time, dosing, etc. As most of the cells presented in this review are not abundantly present in the blood or tissues, in vitro propagation is almost always required to achieve a sufficient cell number for in vivo application. Different biological agents have been used to induce in vitro cell expansion, such as rapamycin [125,126,127]. Rapamycin, also known as sirolimus, is an immunosuppressant routinely used in preventing the rejection of kidney transplants. Interestingly, it has been demonstrated in patients treated with rapamycin that this agent also has a direct in vivo effect on immune-regulatory cells. For instance, rapamycin restored Treg function in six patients with IPEX syndrome treated with rapamycin [314], or induced the upregulation of ILT3 and ILT4 on DCs, thereby promoting the immunoregulatory function of DCs [315]. Similarly, also all-trans retinoid acid (ATRA), which has been used for in vitro expansion of Tregs [129,130], demonstrated a direct in vivo effect on the number of Tregs and IL-10 and FoxP3 expression levels [316]. 

When considering the route of delivery of these cell-based therapies, one needs to consider that different routes lead to different sites of accumulation of the cells administered. Cell-based therapies that can be directed to the lymph nodes and the site of inflammation present an effectual promise of innovative cell-based immunotherapies to battle diseases, such as MS, and to provide a long-lasting cure. In this perspective, we recently presented a novel method to facilitate the migration of cell therapeutic products. Indeed, by introducing messenger RNA (mRNA) encoding CCR5 by means of electroporation (EP), tolDCs transiently displayed increased levels of CCR5 protein expression [304]. Accordingly, the capacity of mRNA electroporated tolDCs to transmigrate toward a chemokine gradient in an in vitro model of the BBB improved significantly, indicative for improved migration of CCR5-expressing tolDCs to inflammatory sites and allowing in situ down-modulation of autoimmune responses in the CNS. In vivo “cell tracking” techniques can pave the way to further optimise current and upcoming cell-based therapies in MS, ranging from preclinical to clinical applications, by improving our understanding of complex mechanisms of action. In vivo bioluminescence imaging allows non-invasive imaging in cell biology and small animal studies [305,306,307,308]. Interestingly, imaging data obtained from mice receiving vitamin D_3_-generated tolDC which were labelled with NIR815 (n = 9), showed that the cells reached the lungs immediately after intravenous administration. Importantly, 24 h after tolDC administration, cells were also found at an elevated concentration in the liver and spleen, up to 7 days post administration [185]. In addition, also in vivo imaging can be deployed in different ways, for instance to stratify patients into responders and non-responders and to predict efficacy or indicate potential loss of efficacy in patients [317]. 

In addition, the heterogeneity in the pathology of MS as well as in its clinical course has presented challenges for the design of therapeutic trials. On top of that, disease heterogeneity has only been partially explained by genetic polymorphisms [318,319,320] and immunological differences in patients [321,322], which can be linked to a higher relapse rate or to a clinical phenotype with more spinal or brain lesions. Hence, well-defined patient selection will account for improved outcome measures. However, currently, there are no biomarkers that adequately predict the individual disease course [323], albeit that some biomarkers, such as neurofilament light, may exert that role in the future [324]. 

Another critical parameter in cell therapy research is the timing when the treatment starts. The window of opportunity for the treatment of patients with MS, directed at downregulating or even silencing the aberrant immune response towards myelin-antigens, is early in the disease course when there is a permeable BBB, a limited amount of axonal damage, before epitope spreading occurs and when the peripheral immune system drives the inflammation in the CNS [325,326,327]. Thus, all cell therapies that intervene with the peripheral-driven immune response should be applied in a timely manner. In addition, targeted cell therapy should ideally be given to patients who show an abnormal T cell response towards these antigens in vitro, which can be found in a subset of MS patients that show inflammatory disease activity [19]. Altogether, the adequate selection of patients for these treatments or for clinical trials is of utmost importance.

Ideally, cell-based therapies must induce increased durability along time. This means that the ability to regulate the autoimmune response must be permanent or at least persist for years following intervention. However, to date, only results demonstrating the safety of tolerance-inducing cell-based therapies in the short-term are available (Figure 2). Indeed, a recent systematic review and meta-analysis evaluating the safety of tolerance-inducing cell-based therapies in autoimmune diseases and transplantation showed that the occurrence of serious adverse events (SAE) is a rare event following treatment with cell-based therapeutic products [328]. Nonetheless, long-term follow-up of participants in well-designed and adequately powered controlled clinical trials is needed to provide evidence of efficacy and long-term safety. 

While it can be hypothesised that reducing the autoreactive, inflammatory assaults in MS may allow for more repair, very little is known about the function of the above-mentioned cell-based therapies in remyelination. Interestingly, since inflammation resolving effects of Tregs frequently occur with tissue regeneration, Dombrowski and colleagues recently revealed a novel proregenerative function for Tregs, as drivers of oligodendrocyte differentiation and remyelination, beyond immunomodulation. This confers a regenerative role for Treg complementary to, but distinct from, known immunomodulatory functions [329,330]. In addition, also MSCs are likely to promote neuroprotection next to their immunomodulatory characteristics [94,95,96], by promoting endogenous repair via local neural precursor cells recruitment. This can possibly be facilitated by the secretion of neurotrophic factors, thereby driving neurogenesis and remyelination [97,98]. These findings will open doors to further optimise cell-based therapies in MS.

Although the first clinical trials reported promising results on the level of safety of administering the cell therapies, discussed in this review, in patients with autoimmune disease in general, and in patients with MS in particular, numerous questions remain unanswered. Ongoing and future studies will help to define the dose, treatment schedule and route of administration of antigen-specific cell therapy in patients with MS regarding safety, efficacy, and treatment-related costs. In conclusion, all aspects of the disease and therapeutic cell product should be considered during cell therapy research, especially within the context of personalised medicine. 

## Figures and Tables

**Figure 1 ijms-22-07536-f001:**
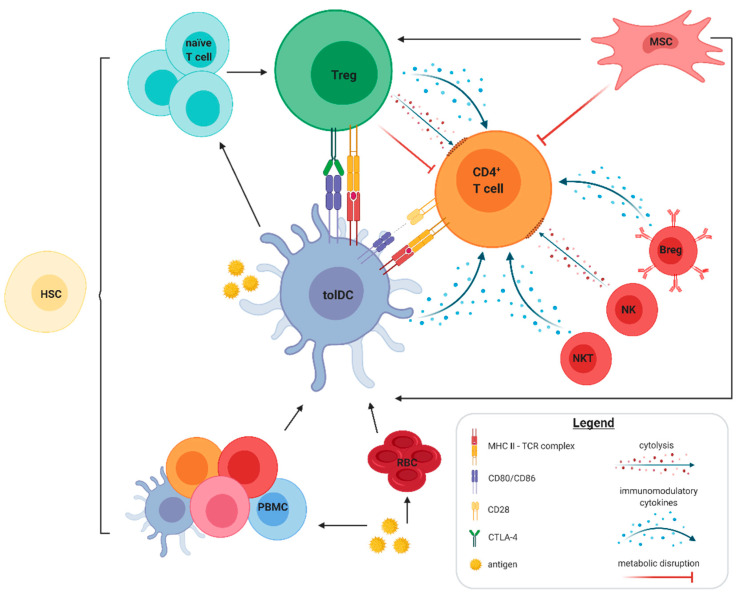
Autologous cell-based tolerance-inducing treatments in multiple sclerosis (MS). This schematic overview depicts the various modes of action implemented by cell-based tolerance-inducing treatments in MS. Additionally shown are the indirect tolerance-inducing strategies by means of inducing a regulatory phenotype in naïve T cells or by peptide-coupled fixed peripheral blood mononuclear cells and erythrocytes. Tolerance induction results in cytolysis, alteration in immune function and/or metabolic disruption of the target autoreactive T cells. The black arrows represent the functional influence. Abbreviations used: Breg: regulatory B cells; CTLA-4: cytotoxic T lymphocyte-associated antigen 4; HSC: haemopoietic stem cells; MHC: major histocompatibility complex; MSC: mesenchymal stromal cells; NKT: natural killer T cells; NK: natural killer cells; PBMC: peripheral blood mononuclear cells; RBC: erythrocytes; Treg: regulatory T cells; TCR: T cell receptor; tolDC: tolerogenic dendritic cells. Created with BioRender.com, accessed date 13 July 2021.

**Figure 2 ijms-22-07536-f002:**
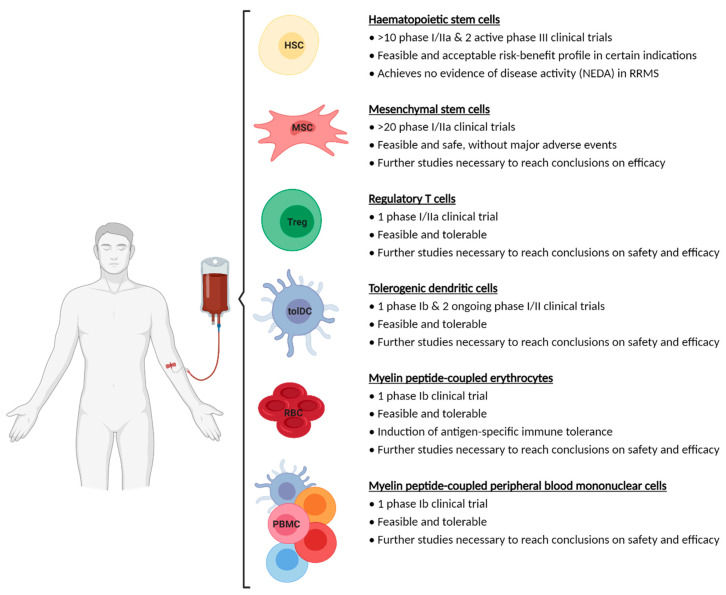
Schematic overview of current cell-based tolerance-inducing treatments for multiple sclerosis (MS) in the clinic. Summary of the results on the feasibility, tolerability, safety and efficiency of cell-based tolerance-inducing treatments that are currently being investigated in different clinical trials is given. Abbreviations used: HSC: haemopoietic stem cells; MSC: mesenchymal stromal cells; PBMC: peripheral blood mononuclear cells; RBC: erythrocytes; Treg: regulatory T cells; tolDC: tolerogenic dendritic cells. Created with BioRender.com, accessed date 13 July 2021.

**Table 1 ijms-22-07536-t001:** Clinical trials in multiple sclerosis (MS) patients. All clinical trials using haematopoietic stem cells (HSC), mesenchymal stem cells (MSC), regulatory T cells (Treg), tolerogenic dendritic cells (tolDC), peptide-coupled peripheral blood mononuclear cells (PBMC) and peptide-coupled erythrocytes (RBC) in MS. A search was conducted in ClinicalTrials.gov on the 16th of June 2021.

	ID	Phase	Design	Status	Cell Type	Route	Administration Scheme	Ref.
**HSC**	NCT00278655	II	Single group assignment, open label	Terminated	Autologous haematopoietic stem cell transplantation	Not provided	Single infusion	N/A
	NCT01099930	II	Single group assignment, open label	Completed	Autologous haematopoietic stem cell transplantation	Intravenous	Single infusion	[40]
	NCT00342134	II	Not provided	Completed	Autologous haematopoietic stem cell transplantation	Intravenous	Single infusion	N/A
	NCT00014755	I	Not provided	Completed	Syngeneic or autologous haematopoietic stem cell transplantation	Not provided	Single infusion	[36]
	NCT00288626	II	Single group assignment, open label	Completed	Autologous haematopoietic stem cell transplantation	Not provided	Single infusion	[41]
	NCT00040482	II	Single group assignment, open label	Completed	Autologous haematopoietic stem cell transplantation	Not provided	Single infusion	N/A
	NCT01679041	II	Single group assignment, open label	Terminated	Autologous haematopoietic stem cell transplantation	Not provided	Single infusion	N/A
	NCT00017628	I	Not provided	Completed	Autologous haematopoietic stem cell transplantation	Not provided	Single infusion	N/A
	NCT00273364	II	Parallel assignment, open label	Completed	Autologous haematopoietic stem cell transplantation	Not provided	Single infusion	[43]
	NCT00497952	I/II	Single group assignment, open label	Active, not recruiting	Allogenic haematopoietic stem cell transplantation	Intravenous	Single infusion	N/A
	NCT02674217	N/A	Single group assignment, open label	Active, enrolling by invitation	Autologous haematopoietic stem cell transplantation	Not provided	Single infusion	[45]
	NCT03113162	I	Single group assignment, open label	Active, recruiting	Autologous haematopoietic stem cell transplantation	Intravenous	Single infusion	N/A
	NCT03477500	III	Parallel assignment, open label	Active, recruiting	Autologous haematopoietic stem cell transplantation	Not provided	Single infusion	N/A
	NCT03342638	III	Parallel assignment, open label	Terminated	Autologous haematopoietic stem cell transplantation	Intravenous	Single infusion	N/A
	NCT04047628	III	Parallel assignment, open label	Active, recruiting	Autologous haematopoietic stem cell transplantation	Not provided	Single infusion	N/A
**MSC**	NCT01377870	I/II	Randomised, double-blind, placebo-controlled	Completed	Autologous bone marrow-derived mesenchymal stem cells	Intravenous	Single infusion	N/A
	NCT02326935	I	Open-label	Terminated	Autologous adipose-derived mesenchymal cells	Intravenous	Single infusion	N/A
	NCT01895439	I/IIa	Open-label	Completed	Autologous bone marrow-derived mesenchymal stem cells	Intrathecal	Not provided	N/A
	NCT02034188	I/II	Open-label	Completed	Umbilical cord-derived mesenchymal stem cells	Intravenous	7 doses	[46]
	NCT01606215	I/II	Placebo-controlled crossover study	Completed	Autologous bone marrow-derived mesenchymal stem cells	Intravenous	Single infusion	[47]
	NCT02035514	I/II	Crossover design	Completed	Autologous bone marrow-derived mesenchymal stem cells	Intravenous	Single infusion	[47]
	NCT01228266	II	Randomised double-blind, placebo-controlled crossover study	Terminated	Autologous mesenchymal stem cell transplantation	Intravenous	Single infusion	[48]
	NCT00395200	I/IIa	Open-label	Completed	Autologous bone marrow-derived mesenchymal stem cells	Intravenous	Single infusion	[49]
	NCT02418351	I/II	Open-label, non-randomised	Terminated	Autologous bone marrow-derived mononuclear stem cells	Intravenous	Single infusion	N/A
	NCT00813969	I	Open-label	Recruitment completed	Autologous mesenchymal stem cell	MSC transplantation	Single infusion	[50]
	NCT02418325	I/II	Open-label, non-randomised	Terminated	Allogeneic human umbilical cord tissue-derived mesenchymal stem cells	Intravenous	Single infusion	N/A
	NCT01056471	I/II	Triple-blind, randomised, placebo-controlled	Recruitment completed	Autologous mesenchymal stem cells from adipose tissue	Intravenous	Single infusion	[51]
	NCT03069170	I	Open-label	Active	Autologous bone marrow-derived mesenchymal stem cells	Intravenous/intrathecal	Single infusion	N/A
	NCT02403947	I//I	Not provided	Active	Autologous mesenchymal stem cell transplantation	Intravenous	Not provided	[47]
	NCT03326505	I/II	Randomised, single-blind	Completed	Umbilical cord-derived mesenchymal stem cells	Intrathecal	Single infusion	[52]
	NCT01745783	I/II	Multicentre, randomised, crossover, double-blind, placebo-controlled	Active, recruiting	Autologous bone marrow-derived mesenchymal stem cells	Intravenous	Not provided	[47]
	NCT02495766	I/II	Randomised, cross-over, placebo-controlled	Completed	Cryopreserved autologous adult bone-marrow mesenchymal stromal cells	Intravenous	Single infusion	N/A
	NCT02239393	II	Randomised, double-blind, cross-over, placebo-controlled	Terminated	Autologous mesenchymal stem cell transplantation	Intravenous	Single infusion	[47]
	NCT01815632	II	Blinded, randomised, cross-over design	Unknown	Autologous bone marrow-derived cellular therapy	Intravenous	Single infusion	[53]
	NCT01854957	I/II	Double-blinded, randomised, cross-over design	Unknown	Autologous mesenchymal stem cells	Intravenous	Single infusion	[47]
	NCT01730547	I/II	Double-blinded, randomised, cross-over design	Unknown	Autologous mesenchymal stromal cells	Intravenous	Not provided	[47]
	NCT02166021	II	Randomised, cross-over, placebo-controlled	Completed	Autologous mesenchymal bone marrow stem cells	Intravenous/intrathecal	Double infusion	[54]
	NCT00781872	I/II	Single group assignment, open label	Completed	Autologous bone marrow derived mesenchymal stem cells	Intravenous/intrathecal	Single infusion	[55]
	NCT01932593	II	Single group assignment, double-blinded	Completed	Autologous bone marrow cells	Intravenous	Reinfusion	[56]
	NCT01364246	I/II	Single group assignment, open label	Unknown	Umbilical cord mesenchymal stem cells	Not provided	Not provided	N/A
**Treg**	EudraCT 2014-004320-22	Ib/IIa	Parallel assignment, open label	Completed	Polyclonal CD4^+^CD25^hi^CD127^−^FoxP3^+^ Tregs	Intravenous/intrathecal	Single infusion	[57]
**tolDC**	NCT02283671	Ib	Single group assignment, open label	Completed	Dexamethasone-tolDC loaded with a pool of myelin peptides	Intravenous	Dose-escalation, 3 injections: bi-weekly	[58]
	NCT02618902	I/IIa	Parallel assignment, open label	Active, not recruiting	VitD3-tolDCs loaded with a pool of myelin peptides	Intradermal	Dose-escalation, 6 injections: 4 bi-weekly and 2 monthly	[59]
	NCT02903537	I/IIa	Parallel assignment, open label	Recruiting	VitD3-tolDCs loaded with a pool of myelin peptides	Intranodal	Dose-escalation, 6 injections: 4 bi-weekly and 2 monthly	[59]
**Peptide-coupled PBMC**	NCT01414634	I/IIa	Parallel assignment, open label	Completed	Myelin-peptide coupled autologous PBMC	Intravenous	Dose-escalation, single infusion	[60]
**Peptide-coupled RBC**	ETIMS^RED^	Ib	Parallel assignment, open label	Completed	Myelin-peptide coupled erythrocytes	Intravenous	Dose-escalation, single infusion	[61]

## Data Availability

Not applicable.

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
