# Peer review of "Made to Measure: Patient-Tailored Treatment of Multiple Sclerosis Using Cell-Based Therapies"

_ijms, 2021, doi:10.3390/ijms22147536_

Round 1
Reviewer 1 Report
The manuscript by Inez et al. summarizes current knowledge on the treatment options for multiple sclerosis with immune system cell-based therapies. In general, the paper is very well prepared and reads well. However, I have only two comments.
- Lines 64-65 The authors should list what are the side effects of DMTs therapy. And just in the context of these side effects highlight the potential benefits of using immune system cells.
- Some cells (e.g. MSCs) require propagation ex-vivo before transfer to the patient. I am curious if any compounds are being used in therapies or ongoing trials that would improve the properties of these cells?
Reviewer 2 Report
Excellent review very well written. Would benefit from proof-reading as a few grammatical errors were evident and the inclusion of figures or table to break up the long text. Some specific points/suggestions for improvement:
- Cell therapy section for the treatment of MS would benefit from a table of all the clinical studies to date with the different cell types.
- Section 2.1 is very comprehensive could you add some detail of the practicalities e.g how much bone marrow or blood can be obtained? is this sufficient for a single or multiple treatments
- In the HSCT section (line 150-154) a discrepancy is highlighted in the literature regarding B cells- can the authors speculate why there is a disagreement?
- In section 2.5.3 what about practicalities wrt NKT cell based therapies can enough of these be isolated and expanded?
- would be great to have an illustration of how antigens can be delivered via cell carriers.
